# IMPROVING MODEL ROBUSTNESS AGAINST NOISE WITH SAFE HAVEN ACTIVATIONS

## ABSTRACT

Quantized neural networks (QNNs) are often used in edge AI because they reduce memory and computational demands. In practical applications such as control systems, medical imaging, and robotics, controlling input noise is crucial for enhancing system robustness. Thus, improving the noise resilience of QNNs is an important challenge in achieving effective edge AI applications. In this paper, we investigate the impact of input noise on QNN performance and propose the *safe haven activation quantization* (SHAQ) method. This approach leverages the characteristics of the quantization function to constrain outputs before quantization within a more noise-resilient 'safe' range, effectively reducing the impact of noise across quantized layers. Our methods achieve state-of-the-art, 73.11% accuracy with 2-bit activations under the fast gradient sign method (FGSM) adversarial attacks with an epsilon of 8/255 on the CIFAR-10 dataset. Furthermore, we extend our methods into a plug-and-play solution we call *quantized helmet* (QH), comprising a series of quantized layers that can be integrated into any unquantized neural network to enhance its noise robustness. Our experimental code and analysis are open-source and publicly accessible.

## 1 INTRODUCTION

*Quantized neural network* (QNN) models use few bits to encode activations of their layers Hubara et al. (2018). This will enable running larger models without requiring more capable hardware and new machine learning capabilities on resource-constrained devices like wearables Yan et al. (2021); Yi et al. (2023); Zhang et al. (2024). Since resource efficiency is crucial in achieving better performance in neural networks, model quantization has been a highly active research area Polino et al. (2018); Fan et al. (2020); He et al. (2024). However, recent research Lin et al. (2023) indicates that basic quantization methods leave QNNs more susceptible to adversarial attacks compared to unquantized models. Therefore, enhancing noise robustness in QNNs without the loss of accuracy in noisy environments is an important challenge.

QANS Lin et al. (2023) and DQ Lin et al. (2019) are previous efforts that incorporated regularization terms based on the Lipschitz constant of each layer's weights to mitigate error amplification in deeper layers due to noisy input. In addition, employing adversarial training strategies, such as using *Projected gradient descent* (PGD) Shafahi et al. (2019); Wong et al. (2020) or adversarial *fast gradient sign method* (FGSM) noise, has enhanced noise robustness Pan et al. (2024); Jia et al. (2024). However, despite using these training strategies, QNN accuracy still suffers significantly under adversarial noise. For instance, when testing a VGG16 model on the CIFAR-10 dataset, the accuracy under clean input with QANS is 90.75%, but it drops to 58.12% under FGSM and 33.82% under PGD-20 ($\varepsilon = 8/255$), reflecting a roughly 30-50% decrease. This drastic degradation in model performance under noisy input highlights the potential for further improvement and is the main motivation of our research.

In this paper, we propose the *safe haven activation quantization* (SHAQ) method. Our method mitigates noise-related performance degradations in QNNs by leveraging quantized activation values. Since quantization functions map a range of input values to a single output value. The key insight is that within this range, some values are more noise-resilient than others. For example, suppose in $[0, 1]$, we quantize it by a step of $0.5$, giving rise to three quantized values, $0, 0.5$, and $1$. In the range of $(0.5, 1]$, the value of $0.75$ can tolerate a noisy input value within $\pm 0.25$ of the original value while

producing the same output value as the quantized activation. Our method aims to push the activation values into such values, we call '*safe havens*' where the greatest amount of noise can be tolerated.

With SHAQ, our quantized neural network achieves a 91.54% accuracy under clean input on the CIFAR-10 dataset using the VGG16 architecture. Under a white-box FGSM adversarial attack with an $\varepsilon$ of 8/255, the model achieves state-of-the-art accuracy (73.11%). Additionally, our method provides superior performance for QNN models against a range of other perturbations, including variations of PGD, random noise, R+FGSM, CW2, and DDN2 Goodfellow et al. (2014); Tramèr et al. (2017); Madry et al. (2017).

Furthermore, we extended our robust QNN training method to enhance the noise robustness of any neural network model. We propose a *quantized helmet* (QH) method where target neural networks are augmented with a learnable 'helmet' structure, which consists of clamped and quantized activations placed at the head of the model. The augmented models are trained with SHAQ, which is designed to penalize input values near the decision boundaries of the clamped and quantized activation functions, pushing these activations into a safer range and thereby minimizing error amplification. Our findings demonstrate substantial improvements in noise robustness across various convolutional neural networks. For example, a non-quantized VGG16 network on the CIFAR-10 dataset, when subjected to an FGSM attack with $\varepsilon$ of 8/255, achieves 21.95% higher accuracy with the 'helmet' compared to without it.

The main contributions of this paper can be summarized as follows:

- We analyze the impact of noise on the performance of QNNs and provide an analytical perspective to understand noise-related performance degradations in QNNs.
- We introduce SHAQ, a training method involving a quantization-aware loss function, to enhance noise robustness by minimizing error propagation. We achieve SOTA accuracy (73.11%) for QNNs under FGSM attacks with $\varepsilon$ of 8/255.
- We propose a plug-and-play 'helmet' structure that consists of clamped and quantized layers. Our Quantized Helmet can be placed at the head of any neural network model, including non-quantized networks. It leverages the same idea in SHAQ but offers a modular solution for improving noise robustness of already existing models.

This paper is organized as follows: Section 2 details our approach to enhancing the noise robustness of quantized models through SHAQ and their application of QH to improve the robustness of unquantized neural networks. In Section 3, we present results demonstrating the improved noise resilience of quantized VGG and ResNet architectures applied to the CIFAR-10 and SVHN dataset, alongside a comparison with existing methods. Finally, Section 4 explores the the combination of our methods with adversarial purification, concluding with a summary of our findings.

## 2 METHODS

### 2.1 SAFE HAVEN ACTIVATION QUANTIZATION (SHAQ)

Quantized activation functions map continuous values into discrete ranges, resulting in information loss that may lead to residual errors. On the other hand, another characteristic of quantized activation functions is that they are stable under perturbations of the input values to a certain extent. Then, a research question arises: Can this natural quantization characteristic be exploited to eliminate the adverse effects of the input noise? Although it is true that most values stay stable under noise, the impact of perturbation is potentially amplified when the perturbed value is near quantization boundaries. Our method proposes a way to tune the network to produce only 'safe' input values that are away from quantization boundaries, eliminating the impact of noise.

We assume the input noise follows a certain symmetrical centered distribution such as uniform distribution $\mathcal{U}[-\varepsilon, \varepsilon]$ where $\varepsilon$ is the scale of the noise.

**Definition 1** (Symmetrical centered distribution). *A distribution is named symmetrical distribution if the probability density function satisfies $pdf(x) = pdf(-x)$ for all $x$.*

We denote $\mathbb{D}$ as the input distribution and the error from the input noise $\delta_x$ as follows:

$$E := \mathbb{E}_{x \sim \mathbb{D}} \mathbb{E}_{\delta_x \sim \mathcal{U}[-\varepsilon, \varepsilon]} |f(x) - f(x + \delta_x)|. \tag{1}$$

We consider the model with the following structure: $f(x) = g(\sigma(Wx + b))$. After the linear layer $Wx + b$, we replace the nonlinear activation $\sigma$ by a floor quantization function to convert activations into $q$-bit representations, where $2^q - 1$ (denoted as $T$) indicates the number of quantization levels:

$$\mathcal{Q}(x) = \frac{1}{T}\lfloor xT \rfloor \tag{2}$$

and assume the later layers to be a continuous function $g(x)$, then we have $f_{\mathcal{Q}}(x) = g(\mathcal{Q}(Wx+b))$. Without loss of generality, we assume $g$ to be Lipschitz continuous with Lipschitz constant $L$. Based on the above Lipschitz continuity assumption, we can have the following characterization of the error from some uniformly random input noise:

**Theorem 1.** *Consider the following quantized model $f_{\mathcal{Q}}(x) = g(\mathcal{Q}(Wx + b))$, we can have the following bound on the error from input noise:*

$$E \leq \mathbb{E}_{x\sim\mathbb{D}}\mathbb{E}_{\delta_x\sim\mathcal{U}[-\varepsilon,\varepsilon]}L|\mathcal{Q}(Wx+b) - \mathcal{Q}(W(x+\delta_x)+b))| \tag{3}$$

$$= L\mathbb{E}_{x\sim\mathbb{D}}\mathbb{E}_{\delta_x\sim\mathcal{U}[-\varepsilon,\varepsilon]}|\mathcal{Q}(Wx+b) - \mathcal{Q}(W(x+\delta_x)+b))|. \tag{4}$$

This theorem indicates a direct relationship between the robustness of individual layers and the overall model's sensitivity to noise and the error can be bounded by $L\mathbb{E}_{x\sim\mathbb{D}}$. By freezing the layer weights in function $g$ and minimizing the error between layers, we can effectively control the total error $E$ caused by input noise. This can be achieved by minimizing the right-hand side of equation 3, which corresponds to the expected distance between the quantized activation values and their noisy counterparts.

We then define the distance function $d_{\mathcal{Q}}$ to be the following input-dependent function $x \in \mathbb{D}$:

$$d_{\mathcal{Q}} = \mathbb{E}_{\delta_x\sim\mathcal{U}[-\varepsilon,\varepsilon]}|\mathcal{Q}(Wx+b) - \mathcal{Q}(W(x+\delta_x)+b)|, \tag{5}$$

We evaluate $d_{\mathcal{Q}}$ using a uniform distribution as a general case since other common noise types, such as random noise, FGSM, and PGD, also produce noise values within the range $[-\varepsilon, \varepsilon]$. The function $d_{\mathcal{Q}}$ is deterministic and depends solely on the quantization method $\mathcal{Q}$. Thus, manipulating the pre-quantization distribution $y = W\mathbb{D} + b$ is an effective way to control the error $E$. In the following theorem, we provide a characterization of the function $d_{\mathcal{Q}}(y)$, which identifies the optimal regions for tuning the distribution of the layer input $W\mathbb{D} + b$.

**Theorem 2.** *There exists a set of safe havens $Y = \{y_i\}$ such that we can represent*

$$d_{\mathcal{Q}}(y) := d_{\mathcal{Q}}\left(y - \arg\min_{y_i\in Y}\ell(y, y_i) + y_0\right). \tag{6}$$

*Here, $\ell$ represents the distance between $y$ and $y_i$, and $y_0$ is the point that minimizes $d_{\mathcal{Q}}$ over the interval $[0, \frac{1}{T}]$. Therefore, minimizing the expected value of the distance function $d_{\mathcal{Q}}(y)$ can be achieved by reducing the expected distance between $W\mathbb{D} + b$ and the safe havens $Y$.*

*Proof.* Since $\mathcal{Q}(y)$ is a piecewise linear function such that, for some integer $T > 0$, $\mathcal{Q}(y + \frac{1}{T}) = \mathcal{Q}(y) + \frac{1}{T}$ for all $y$, we assume without loss of generality that $\frac{1}{T}$ is the period of the function $\mathcal{Q}$. Therefore:

$$|\mathcal{Q}(Wx+b) - \mathcal{Q}(W(x+\delta_x)+b)|$$

$$=|\mathcal{Q}(Wx+b+\frac{1}{T}) - \mathcal{Q}(W(x+\delta_x)+b+\frac{1}{T})|. \tag{7}$$

Taking the expectation over $\delta_y$, and by the definition of $d_{\mathcal{Q}}$, we have $d_{\mathcal{Q}}(y) = d_{\mathcal{Q}}(y + \frac{1}{T})$, where $\frac{1}{T}$ is the period of the function $d_{\mathcal{Q}}$ over the interval $[0, 1]$.

However, the minimal point $y_0$ may not be unique. Although $d_{\mathcal{Q}}$ is piecewise linear, it decreases over $[y_0 - \frac{1}{2T}, y_0]$ and increases over $[y_0, y_0 + \frac{1}{2T}]$. This behavior may not be strictly monotonic, meaning values near $y_0$, such as $y_0 \pm \delta$, could also be safe. When the minimal values are achieved over an interval, we define $y_0$ as the midpoint of this minimal set, as it provides the strongest guarantee of safety by minimizing the risk of deviation. Furthermore, due to the periodic nature of $d_{\mathcal{Q}}$, the minimal values form the set $Y = \{y_0 + k\frac{1}{T}, k \in [T]\}$.

$\square$

We refer to these safe havens as **Q-safety values**, which correspond to the center values of the quantization levels for quantized activations. Thus, based on the previous theorem, we can define the fundamental form of the distance function by:

$$d_{\mathcal{Q}}(y) = \left| \frac{(2\lfloor yT \rfloor + 1)}{2T} - y \right| \tag{8}$$

After applying the quantization function, we further consider the clamp function. In addition to using $\mathcal{Q}$, the clamp function limits the activation function to a fixed range $[c_{\min}, c_{\max}]$, resulting in $q$-bit activations. With the addition of the clamp function, the safe havens need refinement. We introduce a second type of safe havens, called **E-safety values**, to handle edge cases where activation values either fall below $c_{\min}$ or exceed $c_{\max} + 1/(2T)$. These values exhibit enhanced robustness by withstanding larger noise variations, as they will always converge to either $c_{\min}$ or $c_{\max}$ after quantization, regardless of the added noise.

Consequently, the distance function in these cases is zero, reflecting the inherent stability of these values against noise interference:

$$d_{\mathcal{Q}}(y) = 0 \quad \text{if } y < c_{\min} \text{ or } y > c_{\max} + \frac{1}{2T} \tag{9}$$

Additionally, we observe that the interval $y \in [c_{\min}, c_{\min} + 1/2T]$ contains values that lie between $c_{\min}$ and the nearest higher Q-safety point. These values can tolerate more noise due to floor-type quantization, yet under the Q-safety point definition, they would incorrectly be labeled as hazardous. To address this, we propose consolidating these values towards $c_{\min}$ to enhance safety. To maintain the continuity of $d_{\mathcal{Q}}(y)$, we designate all values within $[c_{\min}, c_{\min} + 1/T]$, referred to as Near-$c_{\min}$ safety havens (**NC-safety values**), as equivalent to $y - c_{\min}$, introducing a coefficient $k$ to formalize the distance function accordingly.

$$d_{\mathcal{Q}}(y) = k(y - c_{\min}) \quad \text{if } c_{\min} < y < c_{\min} + \frac{1}{T} \tag{10}$$

Finally, we can define the distance function $d_{\mathcal{Q}}(y)$ as follows:

$$d_{\mathcal{Q}}(y) = \begin{cases} 0 & \text{if } y < c_{\min} \text{ or } y > c_{\max} + \frac{1}{2T} \\ k(y - c_{\min}) & \text{if } c_{\min} < y < c_{\min} + \frac{1}{T} \\ \left| \frac{(2\lfloor yT \rfloor + 1)}{2T} - y \right| & \text{otherwise} \end{cases} \tag{11}$$

An example of the safe havens set and the corresponding distance function for $T = 3$(2-bits), $c_{\min} = 0$, and $c_{\max} = 1$ is illustrated in Figure 1 and Figure 2.

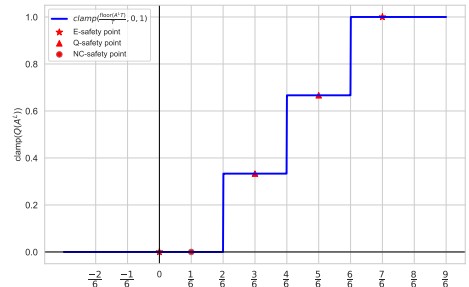
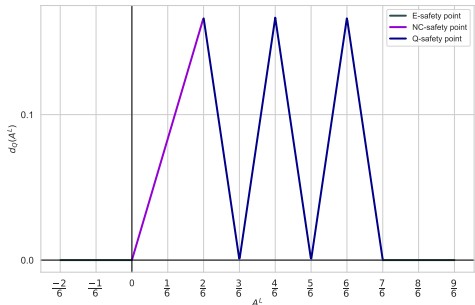

Figure 1: safe haven analysis for $T = 3$ (2-bits activation), $c_{min} = 0$ and $c_{max} = 1$.

Figure 2: Distance function for $T = 3$ (2-bits activation), $c_{min} = 0$ and $c_{max} = 1$.

To enhance the model's resilience, we integrate the sum of $d_{\mathcal{Q}}(x)$ of all layers into the original cross-entropy training loss $L_{ce}$ between output and label as follows:

$$L = L_{ce}(y_{\text{output}}, \text{label}) + \frac{c_1}{2} \sum_{A^L} (d_{\mathcal{Q}}(A^L))^2 \tag{12}$$

This formulation incorporates the distance function to penalize configurations that increase susceptibility to noise, thereby improving the model's overall robustness. We show the final training workflow in Figure 3. The results section will detail how this integration substantively strengthens the model's defense against noise, demonstrating enhancements in robustness.

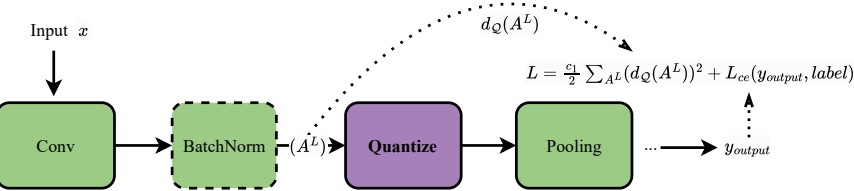

Figure 3: Workflow of SHAQ training.

## 2.2 QUANTIZED HELMET (QH) FOR MODEL ROBUSTNESS

In this section, we aim to use our prior analysis about QNNs to increase the robustness of any network structure, even if it is not quantized. Rather than altering the existing model architecture, we introduce quantization layers that form what we call a *quantized helmet* (QH or $\widehat{H}$) at the front of the original model to receive the input instead of directly feeding it into the model. This strategy leverages the lessons we learned from mitigating noise in QNNs to augment the overall robustness of the (original, unquantized) model.

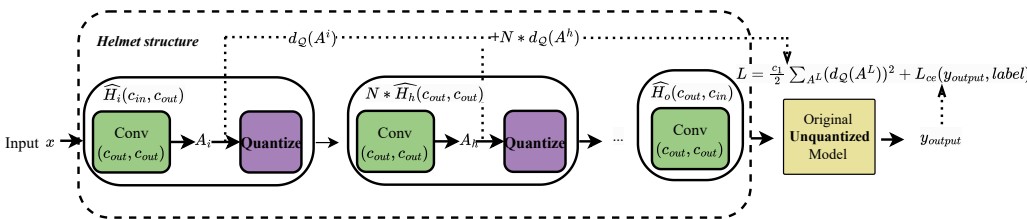

Figure 4: Workflow of quantized helmet (QH).

In our approach, depicted in Figure 4, we preserve the integrity of the target neural network whose robustness we aim to enhance. To this end, we append a 'quantized helmet' including three kinds of quantization layers upstream of the network. The first layer, $\widehat{H}_i(c_{in}, c_{out})$, is configured as a convolutional layer with an input channel count matching the image dimension $c_{in}$ (e.g., 3 for CIFAR-10) and an output channel of $c_{out}$ following by a quantized activation, serving as an encoding layer.

Then, we add $n$ hidden layers $\widehat{H}_h(c_{out}, c_{out})$ which consists one convolutional layer with one quantized activation to help learn the behavior to produce safe haven values. The output $A_i$ and $A_h$ from the initial CNN layer and hidden layers before quantization are preserved to minimize the distance $d_{\mathcal{Q}}(A)$, guiding the data towards a safe haven. This ensures that the output remains unchanged even after adding noise.

Finally, an output convolutional layer, $\widehat{H}_o(c_{out}, c_{in})$, is employed to restore the channel count to the original image size $c_{in}$ before passing it into the neural network, serving as a decoding layer. The loss function for the enhanced architecture, which includes the 'helmet' layers and the neural network, is formulated as the sum of the original network loss and an additional loss, which arises from minimizing the distance $d_{\mathcal{Q}}(x)$. Mathematically,

$$L = L_{ce}(y_{output}, label) + \frac{c_2}{2} \sum_L d_{\mathcal{Q}}(A^L)^2 \tag{13}$$

## 3 RESULTS

### 3.1 EXPERIMENT SETUP

In this study, we utilized CUDA-accelerated PyTorch version 1.12.1+cu116 for training. Experiments were conducted on a system equipped with an AMD EPYC 7763 64-Core Processor, 1000GB of DRAM, and dual NVIDIA A100 GPUs, running Linux 5.15.0-86-generic x86_64.

We choose six kind of adversarial noises in the experiment:

**Random perturbation attack**: This attack involves adding uniformly sampled noise within the range $[-\varepsilon, \varepsilon]$ to the input image. This method does not require any prior knowledge of the data or the network.

**Fast gradient sign method (FGSM)**: Goodfellow et al. (2014) developed the *fast gradient sign method* (FGSM) to create adversarial noise by following the direction of the loss gradient, $\nabla_X L(X, y)$, where $L(X, y)$ denotes the training loss function, such as cross-entropy loss. The adversarial samples are computed as follows:

$$X_{\text{adv}} = X + \varepsilon \cdot \text{sign}(\nabla_X L(X, y)) \tag{14}$$

**R+FGSM**: As FGSM is a single-step, gradient-based method, it may be susceptible to sharp curvatures near data values, potentially resulting in incorrect ascent directions. To overcome this limitation, Tramèr et al. Tramèr et al. (2017) introduced the R+FGSM method, which incorporates an initial random step to move away from these non-smooth regions. The method is defined as follows:

$$X_{\text{adv}} = X' + (\varepsilon - \varepsilon_1) \cdot \text{sign}(\nabla_X L(X, y)), \quad \text{where} \quad X' = X + \varepsilon_1 \cdot \text{sign}(N(0_d, I_d)) \tag{15}$$

In our study, we set $\varepsilon_1$ as $\varepsilon/2$ according to Lin et al. (2019).

**Projected gradient descent (PGD)**: Madry et al. (2017) introduce a more effective variant of FGSM called the projected gradient descent (PGD) method. This method iteratively applies FGSM with a small step size $\alpha$, defined by the equation:

$$X_{\text{adv}}^{t+1} = \text{clip}_\varepsilon \{X_{\text{adv}}^t + \alpha \cdot \text{sign}(\nabla_X L(X_{\text{adv}}^t, y))\} \tag{16}$$

Here, $\text{clip}_\varepsilon(X)$ ensures that the adversarial image remains within the $\varepsilon$-ball around $X$. Same with the approach of Tang & Zhang (2024), we set $\alpha = 0.003$ and iterating 20 times to evaluate model robustness against varying attack strengths.

**Carlini & Wagner (CW) Attack (L2)**: Carlini and Wagner Carlini & Wagner (2017) introduced the CW attack, which differs from gradient-sign methods by minimizing the perturbation while ensuring misclassification. It formulates the attack as an optimization problem with a norm constraint (e.g., $L_2$) and a margin term. Adversarial examples are generated by solving:

$$\min \|\delta\|_2^2 + c \cdot f(X + \delta)$$

where $\delta$ represents the adversarial perturbation, $f(X + \delta)$ is a function promoting misclassification, and $c$ is a constant balancing perturbation size and classification confidence. The $L_2$-norm constrains the perturbation to minimize distortion of the original input. In our implementation, we used torchattacks with a regularization constant $c = 1$, confidence parameter $\kappa = 1$, 10 optimization steps, and a learning rate of 0.01.

**Decoupled direction and norm (DDN)**: Rony et al. Rony et al. (2019) introduced the *decoupled direction and norm* (DDN) attack, which separates the optimization of perturbation direction and magnitude. DDN iteratively adjusts the perturbation's direction based on the gradient of the loss

function, followed by an independent adjustment of the perturbation's magnitude. The direction update is given by:

$$\delta_{t+1} = \frac{\delta_t + \varepsilon \cdot \nabla_X L(X_{\text{adv}}^t, y)}{\|\delta_t + \varepsilon \cdot \nabla_X L(X_{\text{adv}}^t, y)\|}$$

After updating the direction, the perturbation's magnitude $\|\delta_{t+1}\|_2$ is adjusted based on the success of the attack. If the adversarial example is successful, the norm is reduced; otherwise, it is increased:

$$\|\delta_{t+1}\|_2 = \begin{cases} (1 - \gamma) \times \|\delta_t\|_2 & \text{if the attack succeeds} \\ (1 + \gamma) \times \|\delta_t\|_2 & \text{if the attack fails} \end{cases}$$

By decoupling direction and magnitude, DDN efficiently generates adversarial examples with minimal perturbation while maintaining high success rates. In our experiments, we utilized the torchattacks, with parameters set as follows: 20 iterations, $\gamma = 0.05$, initial norm $\|\delta_0\|_2 = 1.0$, quantization to 16 levels, and clipping within the range [0, 1].

## 3.2 EXPERIMENT RESULTS

We present our results for both QNNs and non-QNNs under six different types of noise. The white-box attack results for CIFAR-10 and SVHN using popular architectures such as VGG16 Simonyan & Zisserman (2014) and ResNet-18 He et al. (2016) are shown in Table 1. In QNNs, to prevent input noise errors from growing exponentially, a common strategy involves limiting the network's Lipschitz constant, which is the product of the Lipschitz constants of individual layers. A Lipschitz constant greater than 1, often observed during standard training, indicates that perturbations will be amplified. To mitigate this, a regularization term $\|W^T W - \beta I\|$ is applied, ensuring the orthogonality of the weight matrix rows Lin et al. (2019; 2023). By controlling the Lipschitz constant during quantization, these works effectively limit error amplification. Notably, Lin et al. (2019) and Lin et al. (2023) independently developed *defensive quantization* (DQ) and *quantization adversarial noise suppression* (QANS) to improve model robustness against adversarial noise. For instance, under FGSM attacks with $\varepsilon = 8/255$, DQ and QANS achieved maximum accuracies of 65.52% and 58.12% respectively on CIFAR-10 with VGG-16. In addition this, we also compare our method to non-quantized networks using mainstream adversarial training and image preprocessing methods, such as PGD-AT Madry et al. (2017), TRADES Zhang et al. (2019), and MART Wang et al. (2019). Although these methods achieve higher accuracy under attacks like PGD-20, CW2, and DDN2 compared to QNN approaches, this often comes at the cost of reduced clean accuracy.

In the context of QNNs, our SHAQ method with 2 bits activation setting consistently achieves high accuracy against adversarial attacks while maintaining similar or even higher clean accuracy compared to existing approaches. When comparing our QH method (2-bits 2-hidden-layer quantized helmet) for enhancing noise robustness in unquantized models, it consistently outperforms, achieving 5-10% higher accuracy under clean conditions and against random noise, FGSM, FGSM+R, and DDN2 attacks. For PGD-20 and CW2 attacks, our results are comparable or slightly better than other Non-QNN methods.

## 3.3 ABLATION STUDY

### 3.3.1 QUANTIZED NEURAL NETWORKS WITH SHAQ

We present results for QNNs with activation precision ranging from 2 bits to 4 bits using VGG-16 on CIFAR-10 dataset, both with and without our SHAQ training method, in Table 2, as a case study. Training the entire network for 100 epochs with a learning rate of 0.1, reduced by a factor of 5 at the 30th, 60th, and 90th epochs, we achieve an adversarial accuracy of 60.83%, which is 11.55% higher than the baseline 2-bit QNN. Among all bit levels, the 2-bit configuration provides the highest adversarial accuracy. Since the drop in clean accuracy is minimal and the 2-bit activation leads to a more energy-efficient setup, we select the 2-bit activation as the starting point for further optimization.

Table 1: Comparison with related works.

| Dataset | Methods | | Clean | Random | FGSM | FGSM+R | PGD-20 | CW2 | DDN2 |
|---|---|---|---|---|---|---|---|---|---|
| | | | | | | | Accuracy(%) | | |
| | | | | | VGG-16 | | | | |
| Cifar-10 | QNN | DQ | 90.75 | 90.25 | 65.52 | 77.83 | 41.16 | 53.14 | 30.79 |
| | | QANS | 91.47 | 90.91 | 58.12 | 73.08 | 33.82 | 45.88 | 24.07 |
| | Non-QNN | PGD-AT | 81.11 | / | 51.91 | / | 53.64 | 56.12 | 24.75 |
| | | TRADES | 78.75 | / | 52.84 | / | 56.43 | **66.68** | 29.03 |
| | | MART | 77.79 | / | 54.03 | / | **56.64** | 67.47 | 29.48 |
| | QNN | SHAQ | 91.54 | 90.96 | **73.11** | **78.34** | 56.4 | 66.46 | **54.22** |
| | Non-QNN | QH | **93.52** | **92.88** | 65.39 | 70.79 | 30.06 | 52.89 | 30.21 |
| | | | | | ResNet-18 | | | | |
| | QNN | DQ | 91.18 | 89.32 | 63.00 | 78.15 | 27.26 | 39.44 | 15.92 |
| | | QANS | 90.48 | 89.35 | 66.12 | 78.50 | 17.4 | 28.84 | 11.61 |
| | Non-QNN | PGD-AT | 84.54 | / | 55.11 | / | 48.91 | 59.15 | 19.15 |
| | | TRADES | 83.22 | / | 58.51 | / | **54.07** | 71.62 | 24.24 |
| | | MART | 82.14 | / | 59.57 | / | 54.8 | **74.3** | 25.83 |
| | QNN | SHAQ | 90.54 | 89.20 | 71.56 | 79.82 | 34.82 | 44.72 | 20.54 |
| | Non-QNN | QH | **91.41** | **91.04** | **80.03** | **85.02** | 52.92 | 63.5 | **60.37** |
| SVHN | | | | | VGG-16 | | | | |
| | QNN | DQ | 94.91 | 94.74 | 71.92 | 83.10 | 49.93 | 59.45 | 30.33 |
| | | QANS | 94.67 | 94.63 | 67.74 | 78.96 | 32.10 | 52.31 | 26.79 |
| | Non-QNN | PGD-AT | 92.11 | / | 65.05 | / | 53.64 | 64.15 | 6.6 |
| | | TRADES | 90.83 | / | 66.27 | / | 56.43 | 68.83 | 6.25 |
| | | MART | 92.01 | / | 69.02 | / | 56.64 | 68.78 | 12.04 |
| | QNN | SHAQ | 94.56 | 94.45 | **83.65** | **88.14** | **71.30** | **81.68** | **54.26** |
| | Non-QNN | QH | **95.03** | **94.79** | 79.82 | 84.94 | 53.55 | 66.92 | 40.44 |
| | | | | | ResNet-18 | | | | |
| | QNN | DQ | 94.89 | 94.88 | 79.36 | 85.6 | 22.60 | 35.49 | 19.90 |
| | | QANS | 95.06 | 95.05 | 72.77 | 81.84 | 7.05 | 23.29 | 15.06 |
| | Non-QNN | PGD-AT | 91.66 | / | 87.93 | / | 63.86 | **72.65** | 8.84 |
| | | TRADES | 91.32 | / | 73.38 | / | 59.01 | 72.96 | 5.19 |
| | | MART | 91.81 | / | 75.31 | / | 56.55 | 71.45 | 6.96 |
| | QNN | SHAQ | **95.21** | **95.09** | 80.31 | 86.12 | 44.80 | 55.64 | 24.42 |
| | Non-QNN | QH | 93.42 | 93.26 | **82.76** | **86.94** | **64.23** | 72.34 | **46.70** |

Table 2: VGG16 Clean Accuracy and Accuracy under FGSM attacks ($\varepsilon = 8/255$) on CIFAR-10 with and without SHAQ.

| Methods | Input Type | Full Prec | Quantized Bits | | | Best Accuracy |
|---|---|---|---|---|---|---|
| | | | 2 | 3 | 4 | |
| Baseline | Clean | 93.87 | 92.43 | 93.73 | 93.97 | |
| | FGSM | 42.39 | 51.38 | 38.09 | 27.97 | 51.38 |
| SHAQ | Clean | 93.87 | 92.14 | 92.90 | 92.77 | |
| | FGSM | 42.39 | 60.83 | 60.27 | 60.77 | 60.83 |

Further enhancement was observed when SHAQ (Equation 12) is combined with DQ, which includes an additional regularization term during training. This combination increased the accuracy under FGSM attack to 70.53%.

Finally, incorporating adversarial training ($+Adv$) as suggested by Lin et al. (2019) (additional FGSM+Random with $\varepsilon$ of 8/255 noise to the input data for training), in combination with the SHAQ and DQ training algorithms, results in a clean accuracy of 91.56% and an improvement in FGSM accuracy, reaching 73.11%. This approach provides the highest robustness against FGSM attacks, demonstrating that adversarial training, when combined with other loss functions, is a powerful method for enhancing model robustness.

Table 3: VGG-16 accuracy (%) on CIFAR-10 with SOTA QNN training methods

|  | Full prec | QNN | SHAQ | SHAQ+DQ | SHAQ+DQ+Adv |
|---|---|---|---|---|---|
| Clean | 93.87 | 92.43 | 92.14 | 92.08 | 91.56 |
| FGSM | 42.39 | 51.38 | 60.83 | 70.53 | 73.11 |

### 3.3.2 NON-QUANTIZED MODELS WITH QH

In this section, we compare the performance of the VGG-16 model on the CIFAR-10 dataset with and without the quantized helmet. We begin by presenting an ablation study on the number of hidden layers and their impact on both clean and adversarial FGSM accuracy. We experimented with both 2-bit and 3-bit helmet settings and found that performance did not vary significantly. However, the 2-bit helmet consistently achieved approximately 1% higher FGSM accuracy ($ACC^F$) than the 3-bit setting. Furthermore, it is more energy saving, leading to our choice of the 2-bit activation quantization for the helmet.

As shown in Figure 5 (a) to (d) for the choice of hidden layers, deeper layers capture more robust features against adversarial attacks, increasing FGSM accuracy with depth. However, adding more layers results in a drop in clean accuracy, higher training costs, and increased energy consumption. We found that nearly all robust features were captured with two hidden layers, achieving high $ACC^F$. Therefore, we selected the 2-hidden-layer, 2-bit helmet configuration for all subsequent experiments. For instance, as shown in Table 4, only with QH, although the clean accuracy slightly decreases from 93.87% to 92.15% with the 2-bit, 2-hidden-layer helmet, there is an improvement in accuracy under adversarial FGSM attack, from 42.39% to 64.34%, an improvement of over 20%.

Table 4: VGG16 Clean Accuracy and Accuracy under FGSM attacks ($\varepsilon = 8/255$) on CIFAR-10 with Various Helmet Configurations.

|  | Input type | Without QH | N=1 | N=2 | N=3 | N=4 |
|---|---|---|---|---|---|---|
| 2-bits Helmet | Clean | 93.87 | 92.38 | 92.15 | 91.95 | 91.76 |
|  | FGSM | 42.39 | 64.18 | 64.34 | 65.17 | 65.68 |
| 3-bits Helmet | Clean | 93.87 | 93.17 | 92.92 | 93.06 | 92.5 |
|  | FGSM | 42.39 | 63.56 | 64.25 | 63.68 | 64.43 |

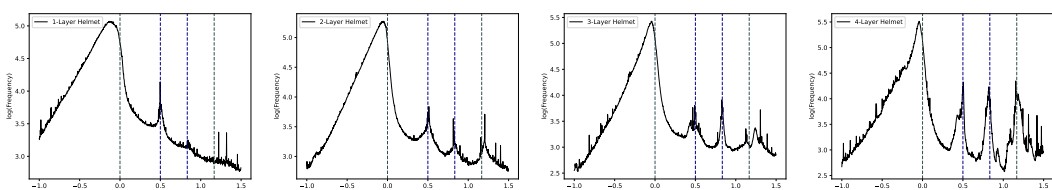

(a) 1-hidden-layer Helmet  (b) 2-hidden-layer Helmet  (c) 3-hidden-layer Helmet  (d) 4-hidden-layer Helmet

Figure 5: Impact of Varying Layer Configurations on 2-bit Helmet Output

## 4 DISCUSSION

In addition to adversarial training, we integrate our method with adversarial purification. Specifically, we adapt the TPAP framework Tang & Zhang (2024), which utilizes an FGSM-robust overfitting network and applies adversarial purification during the testing phase to enhance defense against unknown adversarial attacks. In this process, noise is introduced in the test data, followed by FGSM purification with $\varepsilon = 8/255$, and then tested on the purified noisy data to improve robustness.

As shown in Table 5, combining our algorithm with TPAP results in significant improvements in PGD-20, CW2, and DDN2 accuracies using VGG-16 on CIFAR-10 dataset, which were previously lower when using our method alone, as seen in Table 1. Additionally, compared to using TPAP alone, our combined approach consistently achieves higher accuracy across all noise settings.

Table 5: Cifar-10 accuracy (%) using VGG-16 when combining ours SHAQ with TPAP

|  | Clean | FGSM | PGD-20 | CW2 | DDN2 |
|---|---|---|---|---|---|
| TPAP | 77.02 | 53.4 | 63.99 | 40.57 | 31.2 |
| SHAQ | **90.54** | **71.56** | 34.82 | 44.72 | 20.54 |
| TPAP+SHAQ | 82.43 | 67.64 | **77.9** | **92.09** | **88.59** |

## 5 CONCLUSION

In this paper, we analyze the root cause of quantization-related performance degradations in QNNs. Using the insights that some values in the quantization range tolerate errors better than others, we devised a way to exploit the characteristics of quantized activations to mitigate numerical errors caused by noisy input values. We achieve this by pushing the input values of quantized activations closer to 'safe havens' by fine-tuning the network using an augmented loss function. We introduced our SHAQ method that achieved state-of-the-art accuracy for QNNs under various adversarial noise models on the CIFAR-10 and SVHN dataset with different model architectures, with minimal impact on clean accuracy. Based on the insights of SHAQ, we also introduced the 'plug-and-play' quantized helmet structure that provides a solution to enhance the noise robustness of any neural network model, in particular, unquantized ones. We achieved 88.59% and 92.09% accuracy in the face of the harshest DDN2 and CW2 attacks using VGG16 on the CIFAR-10 dataset. We also showed that SHAQ and QH can complement existing state-of-the-art methods. Future direction of research beyond our work might include conducting a thorough analysis of the trade-offs between model size, inference speed, and robustness when applying our methods to resource-constrained devices, such as mobile phones and IoT devices.

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
