# OpenReview forum: "Improving model robustness against noise with safe haven activations"
_ICLR.cc/2025/Conference — ICLR 2025 Conference Withdrawn Submission_

### Official Review · Reviewer_T5mk · 2024-10-21

**Soundness:** 2
**Presentation:** 1
**Contribution:** 2
**Rating:** 3
**Confidence:** 3

**Summary:**

This paper proposes the Safe Haven Activation Quantization (SHAQ) method for training robust QNN models and extends its application to general neural networks.

**Strengths:**

S1. The paper categorizes the safe havens of QNNs into Q-safety values, E-safety values, and NC-safety values.

S2. The experimental results validate the effectiveness of the proposed SHAQ and Quantized Helmet (QH) methods.

**Weaknesses:**

W1. The presentation in Section 2 is unclear, particularly in the explanation of $d_Q$, which is the most critical component. Specifically, as shown in Eq. (5), it seems that $d_Q$ is treated as a function of $x$, with $\delta_x$ following a uniform distribution. However, in Eq. (6), $d_Q$ is presented as a function of $y$ without any associated distribution. Why does the expectation term $E_{\delta_x}$ disappear in Eq. (6)? Additionally, how are Eq. (5) and Eq. (6) equivalent? The proof does not seem to provide sufficient evidence to support this. Furthermore, I believe it should be $y = Wx + b$ rather than $y = WD + b$, as $y$ represents a value, not a distribution. Due to the issues mentioned above, I became lost starting from Eq. (6) and could only attempt to understand the method through Figures 1 and 2.

W2. I can infer the main point of the paper from Figures 1 and 2, and I believe there is no need to overcomplicate the description of the method. The authors should revise the manuscript to clearly explain the Q-safety, E-safety, and NC-safety values, along with their corresponding $d_Q$.

**Questions:**

Q1. Intuitively, it seems reasonable that QNNs would be more robust than non-QNNs, as quantization can ignore small amounts of noise. However, Table 1 presents conflicting conclusions. For VGG-16 on the CIFAR-10 dataset, QNN-SHAQ demonstrates significantly better robustness than Non-QNN-QH, whereas the opposite is observed for ResNet-18 on CIFAR-10. Similar results are shown for the SVHN dataset. Why is this the case?

Q2. My primary concern when applying QH to general neural networks is the accuracy drop due to the loss of precision (from floating-point to $n$-bit). According to Table 4, adding quantization to VGG-16 does not seem to result in a significant accuracy drop. However, the experimental setup for this table is unclear. When a single hidden quantization layer is added, where is it applied—at a shallow layer or a deep layer? I believe that applying it to a shallow layer would lead to a more pronounced accuracy drop compared to a deep layer. Furthermore, how does applying QH affect other architectures, such as ResNet? The relationship between precision loss and accuracy drop needs further detailed investigation.

---

### Official Review · Reviewer_QD6y · 2024-10-21

**Soundness:** 2
**Presentation:** 2
**Contribution:** 2
**Rating:** 5
**Confidence:** 4

**Summary:**

Quantized neural networks (QNNs) are popular for edge AI due to their reduced memory and computational needs. This paper addresses the challenge of enhancing noise resilience in QNNs, which is critical for applications like medical imaging and robotics. The proposed safe haven activation quantization (SHAQ) method constrains the outputs within a noise-resilient range. Additionally, a modular solution called the quantized helmet (QH) is proposed to integrate quantized layers into unquantized neural networks and enhance their noise robustness. The methods have been tested on various benchmarks and demonstrated model efficiency and robustness in resource-limited environments.

**Strengths:**

1.	The tackled problem is relevant to the community.
2.	The experimental results show that the proposed solution is effective.

**Weaknesses:**

1.	The description of the proposed methods starts immediately in Section 2 after the introduction. It is recommended to add a background section to discuss the necessary background details and extensively discuss the related works with their respective challenges, which provide the necessary motivations for this paper.
2.	It is recommended to discuss the proposed method in more detail through a detailed top-level algorithm describing all the operations involved.
3.	While the experiments have been conducted on various benchmarks. The choice of the benchmarks is questionable. The experiments have been conducted on a combination of relatively old DNN models and adversarial attack methods, and the datasets used CIFAR-10 and SVHN represent a relatively simple classification task. It is recommended to test the proposed method on more challenging benchmarks.
4.	While adversarial training is one of the most powerful adversarial defenses, other defenses have been proposed. It is recommended to compare the proposed method with other adversarial defenses.

**Questions:**

1.	The limitations of the related works should be elaborated in more detail. What are the challenges of existing quantization methods that motivate the need for the proposed safe haven activation quantization (SHAQ) method?

---

### Official Review · Reviewer_aHEr · 2024-11-06

**Soundness:** 2
**Presentation:** 1
**Contribution:** 1
**Rating:** 3
**Confidence:** 5

**Summary:**

This paper introduces a method called Safe Haven Activation Quantization (SHAQ) to improve the robustness of quantized neural networks (QNNs) against input noise, which is crucial for edge AI applications. SHAQ constrains outputs within a noise-resilient "safe" range, significantly enhancing the QNN’s resistance to adversarial attacks, achieving a 73.11% accuracy under FGSM attacks on the CIFAR-10 dataset. The proposed approach is modular and compatible with existing QNN architectures, making it a practical solution for improving noise robustness in neural networks used in real-world applications.

**Strengths:**

The paper specifically addresses noise robustness in QNNs, a unique angle that targets a critical limitation in resource-constrained environments like edge AI devices​. The proposed SHAQ method enhances noise resilience by pushing activations into a noise-tolerant range, achieving notable robustness against various adversarial attacks, such as FGSM, with minimal impact on clean accuracy​.

**Weaknesses:**

-The paper does not include evaluations against stronger, modern adversarial attacks like AutoAttack, which limits the validity of its robustness claims​.

-There is insufficient comparison with recent state-of-the-art defensive strategies, especially non-quantization-based methods, which weakens the context of the proposed method’s effectiveness​.

-Dependency on Simplified Attack Models: The focus on simpler attacks, such as FGSM and PGD, may not fully demonstrate the model’s robustness under more complex real-world adversarial scenarios​.

- The additional QH layers introduce complexity and may increase inference time and energy usage, which is counterproductive for edge AI applications where efficiency is crucial​.

- The experiments are conducted primarily on CIFAR-10 and SVHN, which may not fully represent robustness in more complex datasets or tasks. Broader evaluations could provide stronger evidence of the method's effectiveness​.

- The experiments are limited to relatively small datasets like CIFAR-10 and SVHN, which may not generalize to more complex, large-scale datasets like ImageNet. Testing on larger datasets could strengthen the claims of robustness and applicability in real-world scenarios​.

-The paper exhibits some inconsistencies in writing style and terminology, which may make it harder for readers to follow and fully understand the methodology and results. A more polished and consistent presentation could improve readability and comprehension.

**Questions:**

The paper demonstrates robustness against standard attacks like FGSM and PGD; however, it lacks evaluations against stronger, more modern attacks, such as AutoAttack. Could the authors provide results or analysis on how SHAQ and QH perform against these recent, sophisticated attack methods? Additionally, it would be beneficial to assess the defense under adaptive attacks that are specifically tailored to be aware of the SHAQ and QH methods. This would provide a more comprehensive understanding of the robustness and strengthen the claims made in the paper.

---

### Note · Authors · 2024-11-25

I have read and agree with the venue's withdrawal policy on behalf of myself and my co-authors.